# Luminescent Gold Nanoclusters for Bioimaging: Increasing the Ligand Complexity

**DOI:** 10.3390/nano13040648

**Published:** 2023-02-07

**Authors:** Dario Mordini, Alexandra Mavridi-Printezi, Arianna Menichetti, Andrea Cantelli, Xinke Li, Marco Montalti

**Affiliations:** Department of Chemistry “Giacomo Ciamician”, University of Bologna, Via Selmi 2, 40126 Bologna, Italy

**Keywords:** fluorescence, biocompatibility, photoluminescence, nanoparticles, bioimaging, gold

## Abstract

Fluorescence, and more in general, photoluminescence (PL), presents important advantages for imaging with respect to other diagnostic techniques. In particular, detection methodologies exploiting fluorescence imaging are fast and versatile; make use of low-cost and simple instrumentations; and are taking advantage of newly developed powerful, low-cost, light-based electronic devices, such as light sources and cameras, used in huge market applications, such as civil illumination, computers, and cellular phones. Besides the aforementioned simplicity, fluorescence imaging offers a spatial and temporal resolution that can hardly be achieved with alternative methods. However, the two main limitations of fluorescence imaging for bio-application are still (i) the biological tissue transparency and autofluorescence and (ii) the biocompatibility of the contrast agents. Luminescent gold nanoclusters (AuNCs), if properly designed, combine high biocompatibility with PL in the near-infrared region (NIR), where the biological tissues exhibit higher transparency and negligible autofluorescence. However, the stabilization of these AuNCs requires the use of specific ligands that also affect their PL properties. The nature of the ligand plays a fundamental role in the development and sequential application of PL AuNCs as probes for bioimaging. Considering the importance of this, in this review, the most relevant and recent papers on AuNCs-based bioimaging are presented and discussed highlighting the different functionalities achieved by increasing the complexity of the ligand structure.

## 1. Introduction

The prompt diagnosis of diseases is among the hot topics in the biomedical research field. The preventive detection of malignant conditions in human beings can significantly increase the chances of survival [1,2,3]. About that, the World Health Organization (WHO) demonstrated how the timely diagnosis of tumor and periodic screening can significantly enhance the survival rate in cancer affected people, and it can highly reduce the post-treatment recovery time [4]. In the last years, several investigative techniques, such as computed tomography (CT), positron emission tomography (PET), magnetic resonance imaging (MRI), etc., have been tested as promising diagnostic tools for the detection of internal diseases. Among them, fluorescence imaging has demonstrated encouraging results, such as low-cost performances, high sensibility, and relative safety [5,6,7]. Fluorescence imaging makes use of fluorescence or, more in general, photoluminescence (PL), as a powerful tool for the detection and real-time localization of emitting probes inside a matrix [8]. In the biomedical field, this tool is highly employed especially for the identification of ill sites, for non-invasive endoscopy, and for real-time intraoperative surgery guidance [9,10,11]. It should be noted that fluorescence imaging utilizes PL contrast agents, which, after being injected into the patients, are activated by an external light source, and the imaging is visualized through an optical detector. As a consequence, PL contrast agents need to be highly biocompatible and detectable at a spectral window where biological tissues are more transparent. AuNCs combine these two important features, and they are finding increasing applications in bioimaging the last years. Here we review the most recent examples of this technology on the basis of the level of complexity of the stabilizing ligands that, as is discussed below, affect substantially the properties and the applicability of the AuNCs-based PL contrast agents.

Limited depth of tissue penetration is still a major challenge of the PL probes currently employed in fluorescence imaging [12]. This issue rises from the high absorption coefficient of biological tissues (e.g., skin, fat, etc.) in the visible region [13]. In fact, most of the emitted visible light from the luminescent probes is absorbed by the biological tissues themselves; thus, it does not reach the detector. Such a circumstance lowers the detection limit of the technique, while it negatively affects the analysis due to high light scattering [14]. Moreover, most of the biological matters show autofluorescence, resulting in the reduction of the signal-to-noise ratio. Lately, researchers have explored several ways to overcome this obstacle, and one of the most fruitful approaches is to shift the emission of the employed probes to the near infrared (NIR), working in particular into two specific spectral windows known as NIR I and NIR II. In contrast to visible light, NIR light is more transparent to the biological tissues [15]. Indeed, NIR-emitting probes often show a lower detection limit, lower light scattering, and higher signal-to-noise ratio than the visible-emitting ones [16,17]. Due to that, they guarantee deeper tissue penetration and higher spatial and temporal resolution in the fluorescence bioimaging [18]. Throughout the time, many studies have shown the possibility to design very promising NIR-emitting probes for the detection of endogenous diseases or the display of internal body parts. Organic fluorophores [19,20,21], quantum dots [22,23], and polymeric materials [24,25] were tested as potential luminescent chemicals for applications in the fluorescence imaging. However, in the most recent developments, these emitters have sometimes suffered from certain drawbacks, such as low quantum yield [26], scarce photostability [27], photobleaching [28], or insufficient biocompatibility [29]. AuNCs have been recognized to be more biocompatible than other nanomaterials, while they exhibit acceptable quantum yields in the NIR region [30,31].

AuNCs are typically synthesized upon the reduction of a soluble ionic precursor in the presence of a stabilizing ligand—in many cases, a thiolate—as shown in (Figure 1a) [32]. The presence of the ligands is essential to limit the growth of the metal particles and limit their size to the nanoscale, providing a stable colloidal suspension. Particle size and monodispersity also determine the PL properties. In order to identify gold compounds with a very well-defined chemical composition, the term “nanocluster” is nowadays preferred to nanoparticles. Recently, an increasing number of AuNCs with an atomically controlled structure have been synthetized and characterized [33]. Although not strictly related to this review, we would like to stress that gold functionalization has been exploited to enhance the up-conversion efficiency of other NIR-emitting agents based on lanthanide ions. In one example, over 1000-fold enhancement of up-conversion luminescence was achieved. Nevertheless, in these systems, emission is not originated by the gold clusters [34].

Most characteristic PL properties of AuNCs can be interpreted in terms of molecular orbitals and transitions between electronic states, analogously to what happens in molecules. Both the metal atoms and the stabilizing ligands are involved in the electronic transitions responsible for the PL emission to an extent that depends on their contribution to the HOMO–LUMO. As a consequence, the PL spectrum and quantum yields depend on the gold atom oxidation state and on the nature of the ligands (Figure 1b) [35,36].

**Figure 1 nanomaterials-13-00648-f001:**
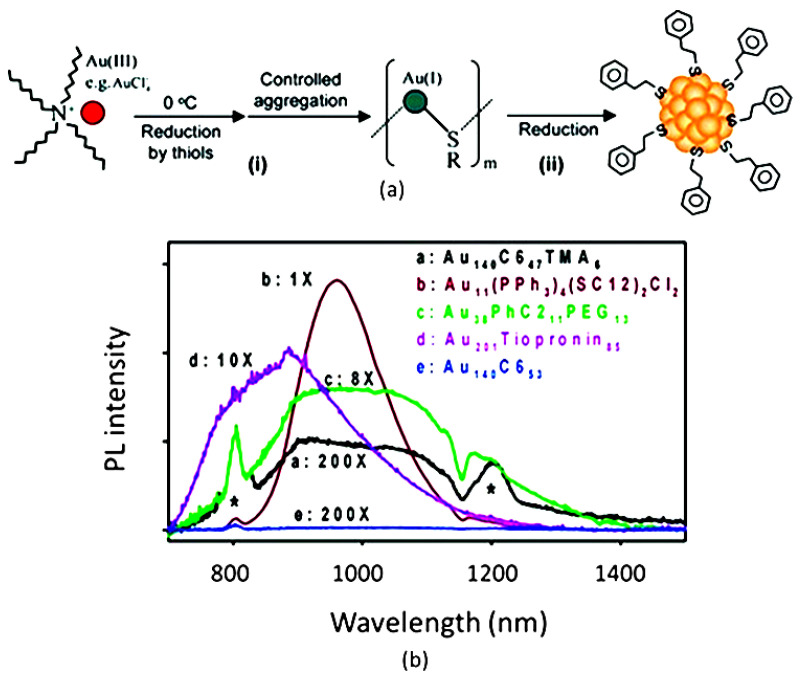
(**a**) Scheme of synthesis of AuNCs. Adapted with permission from Ref. [32]. Copyright 2008 American Chemical Society. (**b**) Comparison between spectra of AuNCs functionalized with different molecules. * Artifacts from second and third-order excitation peaks (800 and 1200 nm). Adapted with permission from Ref. [36]. Copyright 2005 American Chemical Society.

Throughout the years, a large number of different capping agents have been proposed for the development of AuNCs with tailored physical–chemical, photophysical, and biochemical properties [37,38,39,40]. As is shown later in Section 2, the nature of the ligand affects the spectral distribution of the PL allowing, for example, to shift it from the NIR I to the NIR II or even to switch it ON/OFF. Additionally, the ligand can be functionalized for target recognition and therefore drive the accumulation of the AuNCs in specific biological sites. Furthermore, the ligands control the AuNC-AuNC interaction and can be tuned to achieve aggregation/disaggregation. Especially for bioimaging, these supra-particle interactions influence the PL properties and can be exploited for sensing and molecular recognition. Moreover, supra-particle systems can be formed upon AuNCs aggregation, leading to controllable and reversible changes in the PL. Recently, aggregation-induced emission (AIE) and aggregation-enhanced emission (AEE) were reported as one of the most explored strategies to activate or increase the emission of AuNCs tested for a variety of applications, such as sensoring and imaging [41,42].

Considering the importance of the ligands in the development of AuNCs for bioimaging, this paper focuses on the role of the coating agents in the modulation of various photophysical properties, considering increasing ligand complexity, going from thiolated ligands to biobased molecules, and ending with encapsulating coatings. Considering that some review papers about photoluminescent AuNCs have already been published [43,44,45], our discussion is mostly focused on the most recent scientific papers on the topic.

## 2. Design and Application of AuNCs for Bioimaging

The main advantages of the use of AuNCs for bioimaging are (i) the well-documented biocompatibility of these PL contrast agents; and (ii) the light emission in the NIR region, where biological tissues are more transparent and their autofluorescence is negligible. Nevertheless, as discussed above, the nature of the ligand seriously affects both the stability and PL the properties of AuNCs. Considering this important, we decided to base our discussion on the most recent applications of AuNCs for bioimaging, focusing on the complexity of the ligand. In this context, the thiolate-containing ligand plays a dominant, although non-exclusive, role since, due to the well-known affinity of gold for sulfur, they guarantee the necessary good stabilization of the NC.

### 2.1. Low-Molecular-Weight Thiolated Ligands

In this section, we describe characteristic examples of simple low-molecular-weight thiolated ligands, even if the same functionality is also exploited in the biobased approach discussed in Section 2.2. Thiolated AuNCs exhibit large Stoke-shifts and long lifetimes [46,47,48]. Revealing the emission mechanism behind their properties is a fundamental step in designing future technologies based on them. In particular, thiolated AuNCs possess an inner Au(0) core structure and an external shell made of Au(I)-S units [49,50]. The emission of Au(I)-thiol complexes arises from the ligand-to-metal charge transfer (LMCT) and the ligand-to-metal–metal charge transfer (LMMC) on the AuNCs surface, due to Au(I)-S bonds and aurophilic Au(I)-Au(I) interactions [51,52,53]. As is expected, tuning the chemical nature and the number of surface ligands has a big impact on the luminescence features of the thiolated AuNCs. Further studies have demonstrated that the Au(0) core has the main role of stabilizing the Au(I)-S units and sustaining the Au(I)-Au(I) bonding [54,55]. Based on the TD-DFT calculation, the Au(0) core should still have discrete energy states, and because of that, it affects the electronic states of thiolated AuNCs [56,57]. As the research field of luminescent thiolated AuNCs is continuously under development, lately very promising studies were proposed by the scientific community [13,44,45].

In the last years, glutathione (GSH) was reported as one of the most promising thiolated ligands for the functionalization of luminescent AuNCs for imaging applications and sensors [58,59,60]. GSH is a highly biocompatible peptide which is composed of three amino acid units (glutamic acid, cysteine, and glycine), and it plays crucial roles in the physiology of various living beings [61]. In addition to that, the low steric hindrance and the strong binding between the thiol group of the cysteine unit and the Au atoms on the NCs’ surface make it a very stable capping agent [62]. The most common procedure for the synthesis of GSH-capped AuNCs is based on the mixing of HAuCl_4_ and glutathione in aqueous solutions, while after that, the Au(0) precipitation is induced through chemical or thermal reduction, resulting in the self-aggregation of gold atoms into nanoclusters [63,64]. Regarding this, it is well-known that the physical–chemical and photophysical properties of GSH-AuNCs are strictly related to the feeding and the reaction conditions of each experiment [63]. Recently, safer state-of-art synthetic procedures were investigated to create novel GSH-AuNCs.

To begin with, Hada et al. described an innovative microwave-assisted synthetic route for the production of dual-emitting GSH-capped AuNCs [65]. With the new chemical methodology, the formation of the nanoparticles was induced by microwaves, without using any additional hazardous reductant agent, such as NaBH_4_. Novel GSH-AuNCs showed dual-emission in the red and NIR-I regions, at 610 nm and 800 nm, respectively. The former radiative decay had a lifetime of 405 ns, while the latter was 1821 ns long. The origin of the dual emissions was already well described by Jinbin et al. [66]. Briefly, the two emission bands arise from the two possible conformations that GSH molecules can have on the surface of AuNCs and from the number of ligand molecules bonded to the nanocluster. Rescan confocal fluorescence imaging (RCM) and fluorescence lifetime imaging microscopy (FLIM) analysis were tested on GSH-AuNCs-doped tissue-mimicking agarose phantom, showing a good visualization of the probe inside the medium, along with low background noise. The new GSH-AuNCs exhibited long-living emission states, which are crucial in FLIM analysis because they allow for a great distinction between the luminescent probe and the short-living autofluorescence of biological tissues. This study opened the door to a new promising synthetic methodology for dual-emitting GSH-AuNCs in the red and NIR-I region. Furthermore, RCM and FLIM analysis on tissue-mimicking agarose phantom labeled these new GSH-AuNCs as a promising contrast agent for in vitro and in vivo testing. In conclusion, this study demonstrated that GSH-AuNCs with novel features could be produced just by changing the experimental conditions in which they were synthetized, and this should encourage researchers to explore new synthetic methodologies, looking for nanoprobes with novel photophysical and physical–chemical properties.

In the last years, especially in the biomedical field, another innovative approach for the design of new nanotechnologies was pursued. This approach was based on endogenous molecules, which are present inside living organisms and can be exploited for the development of new technologies [67,68]. In particular, the aforementioned nanobiotechnological method is based on the interaction of an artificial probe with a biochemical environment inside a targeted organism. Lately, some research groups have focused their studies on trying to exploit the endogenous GSH for the synthesis of luminescent NCs as efficient probes for bioimaging techniques.

As an example, Peng et al. described a method to synthetize in situ NIR-emitting gold nanoclusters directly inside biological tissues (Figure 2) [69]. Interestingly, the authors showed the possibility of having the site-selective formation of AuNCs, driven by the specific interaction of the capping agents with the biochemical environment of the tissue. More in detail, the synthesis of GSH-AuNCs was performed by placing a section of kidney tissue inside the reaction solution, while the experiment allowed the precipitation of GSH-AuNCs inside the biological matter (Figure 2a,b). Through confocal microscopy, a different quantitative distribution of emitting GSH-AuNC inside the renal tissue was observed (Figure 2c,e,f), and, indeed, most of the nanoprobes were found inside the mitochondria of renal tubule cells (Figure 2d). A possible explanation for these results was proposed: it is well-known that the mitochondria of renal tubule cells act as glutathione storage units; hence, a great absorption of this peptide was expected by this kind of organelle. This phenomenon could be the reason behind the preferential precipitation of GSH-AuNCs inside mitochondria. Additionally, it was confirmed that premade GSH-AuNCs, which were precipitated in solution, did not have the same site-selective targeting efficiency once they were injected into animal models. The experiment was also performed on hippocampus cells, and site-selective precipitation of NIR-emitting GSH-AuNCs was detected, too. Additional experiments using β-glucose thiol as the ligand were carried out in intestine tissue. As was expected, the mineralization of luminescent AuNCs was detected in the brush border, i.e., the major glucose-requiring site of the biological tissue. This suggested that it could be possible to perform highly efficient site-selective in situ synthesis of NIR-emitting AuNCs depending on the chemical nature of the capping agents. Unfortunately, so far, it has been impossible to mimic such reaction conditions in vivo; hence, further paths have to be explored to make this a suitable technology for living beings.

Another interesting approach is based on the knowledge that some classes of cancer cells overexpress glutathione inside the cytoplasm. This interesting phenomenon has also been exploited for the design of novel nanoprobes for bioimaging with highly efficient targeting [70,71].

In view of this, Tan et al. proposed the synthesis of NIR-emitting AuNCs with selective tumor-cell nucleus targeting for bioimaging applications [72]. The authors hypothesized that endogenous glutathione could play a key role in the internalization of these AuNCs inside the nucleus. AuNCs were synthetized using lycosin-l as the ligand. Lycosin-l (L) is a natural peptide which has demonstrated a high affinity for the tumor-cell external membrane, and it is efficient at cancer targeting. Interestingly, it was demonstrated that lycosin-l on AuNCs’ surfaces behaved as an aggregating agent. L-AuNCs self-assembled into supramolecular nanoparticles with a diameter between 40 nm and 80 nm and displayed AEE. In solution, the response of L-AuNCs to the presence of glutathione was verified. L-AuNCs’ photoluminescence decreased when glutathione was added to the system, and a major diminishing was observed when the GSH concentration became higher. From the DLS, it was clear that GSH caused a dis-assemblage of the L-AuNCs’ aggregates, probably because the lower steric hindrance of glutathione makes it a more favorable binder for AuNCs than lycosin-l.

The disaggregation of supramolecular nanoparticles disabled the AEE effect. The uptake of L-AuNCs was tested on malignant cells (4T1 and A549) and non-malignant cells (Hek293t). Thanks to confocal scanning microscopy, it was proven that luminescent L-AuNCs were internalized only by cancer cells. In 4T1 cells, the AuNCs selectively accumulate in the nucleus 8 h after injection. This was attributed to the overexpression of glutathione in tumoral cells, which enhanced the disaggregation of L-AuNCs, allowing the penetration of the disassembled nanoprobes into the nucleus. This study proposed a very promising tool for the fluorescent bioimaging of malignant cells, and it showed how endogenous glutathione, which is naturally present inside the living being, could act as key point in the design of new technologies. This suggested that novel bioimaging tools can be designed through the interaction of capped AuNCs and the biochemical environment of cells.

Except for GSH, in the last few years, several other thiolated agents were tested as suitable ligands for AuNCs in the field of NIR-emitting nanoprobes for bioimaging applications [73,74]. Changing the chemical nature of a ligand expands the number of possible approaches to the problem. Recently, scientists designed multi-capped thiolated AuNCs [75,76,77]. In particular, using more than one ligand at the same time increases the number of variables of the system and makes outcomes more unpredictable. Overall, it allows the design of more sophisticated technologies with superior properties. Here, some of the most recent and representative examples are presented.

To begin with, Pang et al. designed thiolated AuNCs as efficient nanoprobes for the lymph-nodes fluorescence imaging and as excellent drug carrier for chemotherapy [78]. In the current study, NIR-emitting dual-ligand AuNCs were presented as an ideal contrast agent; by tuning the quantitative ratio between the two capping agents, it was possible to reach both good photophysical properties and great site targeting. In vivo, using mice models, a sufficient time of retention was reached. The most relevant example was Au_25_(MUA)_n_(C5)_18−n_ (where MUA is 11-mercaptoundecanoic acid and C5 is a zwitterionic ligand). In NIR fluorescence imaging, Au_25_(C5)_18_-NCs showed great emitting properties, but the lymph-nodes targeting was lacking. At the same time, a large number of MUA molecules bound to the AuNCs were associated with a higher retention time and more efficient site targeting. Indeed, the proposed experiments showed a reduction in the luminescence increasing the quantitative ratio between MUA and C5. The best results were obtained for Au_25_(MUA)_n_(C5)_18−n_, with 4 < n < 9; here, NIR-emitting dual-capping AuNCs were detectable in lymph nodes through fluorescence imaging with a great efficiency. Furthermore, this nanoprobe acted as an efficient drug dealer: methotrexate, a chemotherapeutic drug, was linked to the nanocluster, and the as-prepared system showed greater anticancer properties, such as major tumor-reduction efficiency and less hepatic toxicity. This paper showed how multiple capping could be a way to design thiolated AuNCs with superior features for bioimaging applications and drug delivery. However, it is essential to highlight that changing the quantitative ratio between several ligands could affect, at the same time, multiple other properties. Based on that, deep control of the molecular design in regard to the quantity and the chemical nature of the capping agents is crucial.

Continuing with AuNCs coated with more than one ligand, Yu et al. synthetized a water-soluble shortwave infrared (SWIR)-emitting dual-capped AuNCs with anisotropic surface for the non-invasive bioimaging of vascular vessels (Figure 3) [79]. This study explored the way to enhance the photophysical properties of AuNCs through the selection of the capping agents. It was shown that the addition of tetra (ethylene glycol) dithiol (TDT) on the surface of mercaptohexanoic acid (MHA)-capped AuNCs enhanced the photoluminescent signal up to 12-fold, and it was associated with a redshift in the emission spectrum. The quantum yield for dual-capped (MHA/TDT)-AuNCs was estimated as ~6%, whereas the calculated quantum yield for MHA-AuNCs was 0.9%. The chemical nature of the medium in which (MHA/TDT) AuNCs were dissolved did not have a significant effect: the photoluminescence features of the NCs were unchanged in water and in 10% serum. Surprisingly, in the presence of blood, the photoluminescence increased over time. (MHA/TDT) AuNCs were also tested as contrast agents for SWIR fluorescence bioimaging, and they were proved to possess great properties. Investigation in the SWIR region of the spectrum was associated with a very low tissues autofluorescence, as its transparency to the biological matter is even better than the one in the NIR region (Figure 3a). In in vitro detection of (MHA/TDT) AuNCs, with a concentration of 80 nM, the signal-to-noise ratio was 3.4. This experiment demonstrated that (MHA/TDT) AuNCs displayed clear imaging, with a low concentration of contrast agent inside the target, thus reducing the possibility of toxicity effects. In vivo images were obtained with a great spatial resolution injecting (MHA/TDT) AuNCs into a wild-type mice. To enhance the efficiency of the imaging, computational re-elaboration of the collected pictures was carried out. Post-image processing was performed with the Monte Carlo constrained restoration (MCR) method to further improve the spatial resolution in depth and to overcome the scattering from the tissues (Figure 3b). This technique improved the contrast by 1 order of magnitude and enhanced the spatial resolution by 59%. Image processing was then pushed one step up, using a high-pass filter to reduce scattering deeper under the skin (Figure 3c,d).

Two years later, Le Guével et al. explored the photophysical properties of (MHA/TDT) AuNCs and their application in the bioimaging field more in depth [80]. In particular, this study concerned the effect of solvent polarity on the photoluminescence properties of dual-capped (MHA/TDT) AuNCs and how the rigidity of the matrix, in which the NCs were dispersed, affects their emission. The authors analyzed several emission spectra of (MHA/TDT) AuNCs solubilized in various mixtures of DMSO and water at different molar ratios, from 0 to 90%, of DMSO. DMSO was chosen as co-solvent due to the fact that it is less polar than water, it is miscible with aqueous medium, and it guarantees a good solubility of (MHA/TDT) AuNCs. The emission of (MHA/TDT) AuNCs was characterized by three emission peaks that are namely at 920 nm, 1050 nm, and at 1200 nm. It was highlighted that the relative enhance in intensity of the peak at 1200 nm with respect to the peak at 920 nm was proportional to the increase of molar-ratio percentage of DMSO in the samples. It was hypothesized that this effect could be dependent on the charge-transfer efficiency on the differences between the dielectric constant of the medium and the ligand shell. This information suggested that the polarity of the medium could be a parameter that is able to tune the intensity of emission of (MHA/TDT) AuNCs in the SWIR region for a more efficient bioimaging analysis. Furthermore, it was investigated how the rigidity of the matrix in which (MHA/TDT) AuNCs were dispersed affects the photoluminescence of the NCs. Interestingly, a progressive redshift to 1450 nm was observed according to a gradual increase in the rigidity of the matrix. As a future perspective, maybe it will be possible to tune the wavelength of emission of (MHA/TDT) AuNCs just by optimizing the rigidity of an encapsulating agent around the NCs, something that would be very beneficial in the fluorescence bioimaging field. Finally, (MHA/TDT) AuNCs were tested as a contrast agent for the fluorescent bioimaging of a 3D reconstruction of the ventral blood vessel. (MHA/TDT) AuNCs were demonstrated to be suitable nanoprobes for the nice visualization of the vascular network at different angles and at different depths, while the 3D structure was built using an IterNet neural network. From this study, it is clear that SWIR-emitting (MHA/TDT) AuNCs could serve as promising contrast agents for the visualization of blood vessels. Additionally, it was demonstrated that they possess easily tunable photophysical properties. In the future, it will be highly advantageous to explore the possible selective organ-targeting using (MHA/TDT) AuNCs and to have the chance to create theragnostic devices, which are among the new frontiers of biomedical research.

### 2.2. Biobased Ligands

In the few last years, biomolecules have caught the attention of the scientific community as promising capping agents for metallic nanoparticles (MNPs). The main advantage of this approach is that these ligands may increase the biocompatibility of the metal-based probes. The emergent need to find biocompatible, non-toxic, and green chemicals for the synthesis of nanobiomaterials has driven researchers to test natural molecules as ligands for MNPs [81,82]. Among them, proteins [83,84], peptides [85,86], amino acids [87], carbohydrates [88,89], and plant extracts [90,91] have been demonstrated to have a big impact on the biochemical, photophysical, and physical–chemical properties of MNPs. Moreover, recently, AuNCs capped with biomolecules showing superior properties have been reported in the literature.

The large number of available amino acids offers the chance to develop well-designed capping agents for AuNCs, selecting the ones with the desired physical–chemical properties. For example, thiolated ones were exploited as reductants and stabilizers in the synthesis of Au(I)-S complexes, while aromatic ones can tune the photophysical features of the emitting AuNCs [92,93]. Amino acids can condensate into bigger molecules, such as peptides and proteins. These macromolecules possess 3D structures and high biochemical activity, which can contribute to the design of capped AuNCs with higher-level performances. The most popular protein used as a capping agent for AuNCs is bovine serum albumin (BSA) due to its high availability, relatively low cost, and the presence of several thiolated units which help in the anchoring of AuNCs inside its structure [94,95,96]. Bertorelle et al. hypothesized that BSA is able to enhance the quantum yield and induce a redshift as a capping agent for AuNCs. This was due to the fact that the local environment of AuNCs attached to BSA was more constrained and prone to multiple energy transfer associated with intersystem crossing. Moreover, the capability to interact through electron-rich donor groups with the surface of the AuNCs core may increase the number of surface states involved in excitation [97]. Lately, several examples of BSA-capped AuNCs were described; for example, Dan et al. described an innovative NIR-II-emitting BSA-capped AuNCs (BSA-AuNCs) with catalase-like activity for the photodynamic treatment of cancer and the healing of MRSA-infected wounds [98]. In this paper, the experimental conditions and the feeding of the synthesis were well studied, and it was highlighted that the photoluminescence features of BSA-AuNCs were strictly related to the chosen to set up. It was noticed that the NIR-II emission of BSA-AuNCs was significantly dependent on the molar ratios of the reactants. Indeed, the best results in terms of photoluminescence intensity were obtained for BSA:HAuCl_4_ and NaOH:NaBH_4_ equal to 0.15 and 25, respectively. In addition, the authors evaluated the effects of the reaction time on the fluorescence of BSA-AuNCs, and it was reported that the brightness of BSA-AuNCs increased in time during the reaction. The luminescence increased until 12 h after the beginning of the synthesis; meanwhile, later, a lowering in the luminescence of the BSA-AuNCs was observed. BSA-AuNCs, which were synthetized for 12 h at 37 °C, displayed a calculated quantum yield of around 3.5%. As it is reported, the chemical reduction of HAuCl_4_ by using NaBH_4_ was an essential step to achieve the emission in the NIR-II region. The fluorescence spectra of BSA-AuNCs were recorded under various excitation wavelengths, in the range of 450 nm to 808 nm, and an excitation wavelength-dependent photoluminescence was shown. The NCs displayed two emission bands, with the first being in the NIR-I region (960 nm) and the second being in the NIR-II region (1010 nm). In vivo, the great photophysical properties of the BSA-AuNCs were exploited to perform NIR fluorescence imaging of tumor sites in mice models. Thus, NIR-II fluorescence imaging showed a great resolution of the tumor site compared to the NIR-I fluorescence imaging, due to the lower absorption and photon scattering of biological tissues. The NIR-II tumor fluorescence signal was about 7.3-fold higher than that of the background at 10 h post-injection, with great liver and tumor accumulation. Indeed, BSA-AuNCs were demonstrated to be great agents for photodynamic therapy, both in tumor killing and wound disinfection. The developed nano-agent also displayed good stability in several media, photobleaching resistance, biocompatibility, low toxicity, and deep penetration detection. On the whole, BSA was proved to be an excellent capping agent for AuNCs, with great theragnostic features. This suggests that natural proteins could be excellent ligands, both in the tuning of photophysical features and for selective site targeting.

Other proteins have also been tested as capping agents for AuNCs. The broad range of available proteins opens the door to an almost infinite number of possible outputs.

For example, Wang et al. presented NIR-II-emitting Ribonuclease (RNase)-capped AuNCs (RNase-AuNCs) for the imaging of the gastrointestinal tract and the detection of cancers in the intestine (Figure 4) [99]. One of major issues in the imaging of digestive system is the very harsh conditions of the environment: acidic juices and enzyme-rich sectors tend to negatively affect the properties of standard probes. From the DFT calculation, it was hypothesized that aromatic amino acids (histidine and tyrosine) that are bound on the AuNCs surface could promote a redshift in the protein-capped AuNCs, thus reducing the Fermi energy of AuNCs (Figure 4A). Because of that, RNase protein corona was engineered with eight cysteine, four histidine, and six tyrosine units. As in the case of other biobased ligands, the thiolated amino acid worked as a stronger linker between AuNCs and the protein, while the aromatic amino acids affected the LMMCT mechanism of RNase-AuNCs, promoting a redshift (Figure 4B). These novel RNase-AuNCs emitted in the NIR-II region with a quite narrow emission peak centered at 1050 nm, with a calculated quantum yield of around 1.9%, which is higher than the one of the most common NIR-II emitters. RNase-AuNCs demonstrated good stability in simulated gastric juice and great biocompatibility. NIR-II fluorescence imaging of the gastrointestinal tract was performed by oral injection of the RNase-AuNCs in animal models (Figure 4C). A dynamic movement process of AuNCs was observed in gastrointestinal peristalsis, including stomach duodenum, ileum, and caecum, with a high resolution of 1.6 mm, which is 4-fold higher than that of barium swallow. It was possible to detect a tumor of 2.5 mm in diameter in the intestine of an oncologic mice. Overall, these new RNase-AuNCs demonstrated that it is possible to design AuNCs with tailored features through the functionalization of proteins. The key point was to elucidate how the binding of several amino acids to the AuNCs’ surface affect the photophysical properties of the nanoprobes. This can encourage scientists to understand, in depth, the photophysical mechanisms behind the emission of contrast agents in order to design materials with desired characteristics.

Natural proteins demonstrated great performances as capping agents for AuNCs; however, it was reported that synthetic peptides could also be promising capping agents to tune the photophysical properties of AuNCs [100,101,102]. Indeed, artificial peptides have the big advantage of being entirely tailorable in terms of their chemical structure. This could be very helpful for the design of capped AuNCs with well-defined features. An interesting study was the one proposed by He et al., who showed an NIR-I-emitting novel cyclopeptide (CP)-capped AuNC (CP-AuNCs) with promising gene-delivery features and tumor-site imaging [103]. The authors projected a synthetic cyclopeptide, cyclo-(Ala-Arg-Ala-Arg-Ala-Arg-Ala-Asp)-aminocaproic-Cys, which could spontaneously self-assemble into nanofibers, which were exploited as a template for the production of 1D nanostructures made of AuNCs. The desired result was obtained by performing the reaction in weak alkaline conditions, using the CP as a template in the presence of HAuCl_4_ and glutathione. The photoluminescence properties of CP-AuNCs showed a strong emission peak at 810 nm and a quantum yield around 5.9%, almost 6-fold higher than the standard NIR-emitting AuNCs. Related to that, these novel CP-AuNCs exhibited great photostability in physiological medium and in high-ionic-strength solutions. In opposition to GSH-AuNCs, CP-AuNCs were able to be incorporated by living cells through endocytosis, escape from lysosomes rapidly, and be excreted by the cells fast. The aforementioned properties clearly labeled these nanoprobes as being low in toxicity and high in gene-transfection efficiency. In accordance with that, CP-AuNCs acted as good nanocarriers for important tumor suppressor genes inside malignant cells. Fluorescence NIR-I imaging was used for the visualization of the internalization process of CP-AuNCs, as this is a key step to ensure the efficiency of nanocarriers. In conclusion, this paper was very promising, as it demonstrated that the amino acid sequence of synthetic CP can be designed to obtain capping agents with well-designed features. It was also shown that photoluminescence imaging can be a promising tool to study the pharmacodynamic and to evaluate the drug fate.

Another big family of biomolecules is carbohydrates. β-cyclodextrins are one of the most promising polysaccharides able to act as a capping agent for AuNCs, as they guarantee excellent host–guest recognition, high biocompatibility, and good solubility in aqueous media, such as body fluids [104]. In view of this, β-cyclodextrin-capped AuNCs (CD-AuNCs) have also been demonstrated to be promising emitting probes both in sensoring and imaging. According to Wang et al., red-emitting β-cyclodextrin-capped GSH-AuNCs (CD@GSH-AuNCs) showed more intense photoluminescence compared to nude GSH-AuNCs [105]. The authors proposed that the rise in emission properties was due to the presence of a major number of electron-rich atoms, such as oxygen, on the AuNCs surface, as this would induce a ligand-to-metal cluster-core charge transfer and a decrease in the intramolecular rotation. Nonetheless, until now, not many NIR-emitting CD-AuNCs have been reported in the literature. One of the few examples was presented by Song et al., who proposed NIR-II-emitting β-cyclodextrin (CD)-capped AuNCs (CD-AuNCs) for the labeling of proteins and antibodies as a potential tool for site-selective fluorescence imaging [106]. The authors synthetized CD-AuNCs through the reduction of HAuCl_4_ in the presence of mercapto-β-cyclodextrin. These CD-AuNCs showed an emission peak at 1050 nm upon 808 nm excitation, with a quantum yield of 0.11%. By lowering the temperature, it was possible to observe an increase in the intensity of luminescence and a longer lifetime. These novel CD-AuNCs also showed good stability in different media, poor cytotoxicity, and high renal cleanability. Moreover, it was shown how CD-AuNCs can label proteins, such as BSA (BSA@CD-AuNCs), or antibodies, such as tumor-targeting antibodies (Ab@CD-AuNCs). Through NIR-II fluorescence imaging, it was demonstrated that the labeling with a protein or an antibody can tune the site-targeting and retention properties of CD-AuNCs, showing a 3-fold enhancement of the NIR-II signal intensity compared to the control group with a penetration depth of around 9 mm. Importantly this study presented a very promising tool for in vivo imaging, as the main advantage of these nanoprobes is the possibility to select a target just by selecting the appropriate label, making it a very versatile tool.

Nucleic acids represent another big class of biomolecules which are extensively exploited in the field of nanobiomaterials. Nanotechnologies based on DNA strands have become highly demanding, as they can provide high-level tasks such as gene therapy and DNA-target-recognition techniques [107,108]. Recently, DNA was also tested as a promising ligand for AuNCs.

Dai et al. described the synthesis of AuNCs, which were functionalized with phosphorothioates (ps)-modified DNA (psDNA) for fluorescent optical imaging in the NIR region (Figure 5) [109]. The psDNA-capped AuNCs (psDNA-AuNCs) were obtained by adding psDNA into the glutathione-Au(I) complexes solution formed by a mixture of HAuCl_4_ and glutathione. Until now, the synthesis of DNA-capped ultrasmall AuNCs was challenging for several reasons, such as the large electrostatic repulsion of AuNCs, the restricted area on AuNCs surface, and the difficulty in the capping-agent exchange between DNA and glutathione. Herein, the employed DNA strands were designed in a very smart way: one of the extremities of the selected DNA string was functionalized with ps-nucleotides to induce the formation of strong Au(I)-S bonds on the surface of AuNCs (Figure 5a). It was reported that the number of ps-nucleotides at the extremities of the selected DNA strands affects the reaction yield. For AuNCs functionalized with one DNA string, the greatest amount of product was obtained when ps-nucleotides were seven, while for AuNCs bound to two DNA strands, the highest reaction yield was obtained for five ps-nucleotides. This is due to the fact that the number of ps-nucleotides affects both the steric hindrance and the reduction potential (Figure 5b). The photoluminescence of psDNA-AuNCs exhibited a strong NIR-I emission with a peak at 810 nm. For live-cell fluorescent imaging, the psDNA-AuNCs were hybridized with the sgc8c aptamer (Apt) that targets PTK7 proteins overexpressed on the membranes of CCRF-CEM cells (Figure 5c). A high fluorescence signal was obtained from the CCRF-CEM after incubation with Apt@psDNA-AuNCs, and in addition to that, high cell-type selective targeting was achieved (Figure 5d). These results clearly demonstrated that gold nanoparticles can be biocompatible luminescent nanoprobes in the design of DNA-based nanotechnologies.

### 2.3. Encapsulating Coatings

In this section, we discuss how, by using encapsulating coatings, it is possible to achieve new functions and new responsive materials with switchable properties. As was described so far, the chemical bond between a molecule and the surface of AuNCs can dramatically rearrange the electronic states of the nanoparticles. Indeed, heteroatoms bound on the surface of AuNCs can strongly perturb their photophysical properties [110,111]. In addition, other strategies were explored in order to design AuNCs with suitable physical–chemical properties for application in the fluorescence imaging. The encapsulation on luminescent nanoprobes showed promising results in the modulation of the photophysical properties of the nanoprobes [112,113]. In fact, coatings can have a big impact on the aggregation state of AuNCs, resulting in different interparticle interaction and a different response to the external environment and stimuli [114,115].

In this context, the chemical coating around AuNCs can play a crucial role. In reality, several cappings were reported as efficient agents for the promotion of NPs’ aggregation or NPs’ disaggregation. Recently, AIE and AEE were reported as one of the most explored strategies to activate or increase the emission of AuNCs for a variety of applications, such as sensoring and imaging [41,42]. It was hypothesized that AIE and AEE were promoted by the blocking of intramolecular vibrational and rotational relaxation pathways, promoting radiative emission mechanisms [116,117,118]. In recent years, the AEE mechanism was exploited for the design of NIR-emitting AuNCs-based nanoprobes for fluorescence imaging in the biomedical field.

To begin with, Tang et al. proposed a NIR-II-emitting nanoprobe for intestinal bioimaging based on AuNCs coated with an amphiphilic block copolymer as both micelles and unimers [119]. Pluronic F-127 is an amphiphilic copolymer that is composed of a hydrophobic central poly(propylene oxide) (PO) block and two lateral hydrophilic poly(ethyl glycol) (PEG) blocks. Beyond the critical micelle concentration (CMC), Pluronic F-127 self-assembles into micelle with a hydrophobic core and hydrophilic surface. Both pluronic F-127 unimers and micelles were demonstrated to serve as a good template for the synthesis of AuNCs coated with ethyl 3-mercaptopropanoate as a ligand (EMP-AuNCs). The unimeric AuNCs showed a red-shifted emission with a maximum at 1280 nm, a quantum yield of 0.25%, an emission lifetime of 68 ns, and 33.3% of Au(I) ions in the NCs, while it was demonstrated that micelles were loaded with nine AuNCs, and they showed an emission band with a maximum peak at 1080 nm, a quantum yield of 1.6%, an emission lifetime of 200 ns, and 48.5% of Au(I) ions per AuNCs. In the former case, it was hypothesized that the lower energy emission and the shorter lifetime were due to the enhanced steric hindrance and the interparticle hydrophobic interaction, which decreased the emission energy levels from the Au core states, while for the latter case, the micelles’ nanostructures favored the ligand-to-metal charge transfer (LMCT) processes more than the unimer structures, so longer and more energetic emissions were promoted. The nanoprobes demonstrated good stability in gastrointestinal-like juices, and they were orally injected into mice with ulcerative colitis. Strong luminescence and great visualization of the intestine region were successfully achieved for both kinds of nanoprobes. Interestingly, unimeric AuNCs were retained in the gastrointestinal tract for a longer time in respect to micelles loaded with AuNCs, something that demonstrated that unimeric AuNCs had a stronger interaction with the injured intestinal mucosa. This study was very interesting because it highlighted how two systems with the same chemical nature can have crucial differences in their photophysical properties and in their affinity for biological tissues due to the different aggregation state.

Another study by Zhu. et al. described an NIR-emitting nanostructure based on Ag(I)-doped GSH-AuNCs encapsulated with fluorinated polymer (PF) with a high quantum yield as a multimodal nanoprobe for in vitro and in vivo imaging [120]. GSH-AuNCs were mineralized inside cysteamine-grafted poly(ethylene-alt-malic anhydride) functionalized with 2,2,2-trifluoroethylamine as a template agent. PF@GSH-Ag(I)AuNCs exhibited an emission in the NIR region, with a maximum at 810 nm and a quantum yield of 27.7%, which was very high compared to the other luminescent AuNCs-based nanoprobes. Ag(I) played a key role in the modification of the photophysical properties of AuNCs increasing the photoluminescent intensity, and it induced a redshift in the emission. In parallel, the increasing PF concentration in the feeding of the reaction was associated with a blueshift in the emission, although no emission from the nanoprobes was observed when PF was added below a threshold concentration limit. Both GSH and the alkaline reaction environment promoted a redshift in the PF@GSH-Ag(I)AuNCs emission. NIR-emitting PF@GSH-Ag(I)AuNCs were tested on 3D 4T1 tumor spheroids for fluorescence imaging, and as was expected, fluorine guaranteed higher permeability through the cellular membrane due to low-energy surface and lipophilic properties in respect to the halogen-free coating. It was further demonstrated that PF@GSH-Ag(I)AuNCs was able to aggregate in weak acid solution, something that was associated with an increase in the emission intensity due to the restriction of the non-radiative relaxation of the ligand vibration, solvent relaxation, and internal conversion. Indeed, PF@GSH-Ag(I)AuNCs accumulation was found inside the cellular lysosomes, which have an acid internal microenvironment. On top of that, PF@GSH-Ag(I)AuNCs were tested on Balb/C mice, where a great visualization of the tumor site was observed through fluorescence bioimaging; in addition, multimodal imaging was performed with ^19^F MRI and CT.

AIE and AEE are not the only phenomena observed when multiple NCs are aggregated in a constrained space. Recently, some studies reported the encapsulation of a large number of emitting NCs in a confined space, which promoted a lowering of the emission quantum yield and of the emission lifetime. It was observed that aggregates of the same NCs can reduce the photoluminescence of the nanostructure. This phenomenon was considered a promising tool in the design of disassembly-induced enhanced emission (DIEE) [121,122].

In view of this, Zhou et al. proposed a pH-induced self-assembly of NIR-emitting AuNCs for fluorescence bioimaging based on the DIEE (Figure 6) [123]. The presented AuNCs were coated with 3-mercaptopropyltrimethylsilane (MPTMS) and a thiolated PEG (Figure 6a). Trialkoxysilane was selected, as it can induce the condensation of alkoxysilane groups and the formation of cross-linking points, such as Si-O-Si bridges, among the AuNCs, which promotes the AuNCs’ aggregation, while the hydrophilic PEG can enhance the solubility of the nanoprobes inside water-rich biological tissues. As it was demonstrated, the degree of cross-linking was dependent on the pH of the medium: acid environment promoted a strong hydrolysis of Si-O-Si bonds, whereas in basic media, the Si-O-Si bridges are quite preserved (Figure 6b). Disaggregation was associated with a blueshift, a longer lifetime, and a higher quantum yield. MPTMS/PEG-AuNCs in aqueous medium at pH 4.0 showed an emission with a maximum at 955 nm, a lifetime of 960 ns, and a quantum yield of 12%, while MPTMS/PEG-AuNCs in aqueous medium at pH 8.5 showed an emission band with a maximum at 1070 nm, a lifetime of 187 ns, and a quantum yield of 1.8%. It was hypothesized that the formation of Si-O-Si between the AuNCs hampered the ligand-to-metal charge transfer processes, and it was associated with an enhancement in the emission pathways of the NCs core and a quenching of emissions from the NCs’ surface. Both assembled and disassembled nanoprobes were tested for in vivo fluorescence imaging. The former was more retained inside the mice body than the latter; however, both selectively accumulated in the liver and in the spleen and displayed good imaging of the organs (Figure 6c).

Exploiting another approach, the emission quenching can also be exploited to promote shifts in the AuNCs emission.

Haye et al. designed highly bright NIR-II-emitting nanoprobes composed by a high amount of AuNCs encapsulated in small polymer nanoparticles for imaging in tissue-mimicking models [124]. Up to 14,000 Au_25_DDT_18_ (DDT = dodecanethiol) NCs were encapsulated inside a single poly(ehtyl methacrylate) (PEM) NP, using the supersaturation nanoprecipitation synthetic approach. The encapsulated AuNCs exhibited a broad emission band in both NIR-I and NIR-II regions, with a maximum around 1030 nm. Increasing the concentration of AuNCs inside the polymer NP was associated with a redshift in the emission and an enhancing in absorbance. In addition, a lowering of the quantum yield was observed. Regardless, the absolute photoluminescence intensity of NCs still increased, showing that the increasing number of NCs inside NPs significantly compensated for the decrease in the quantum yield. Related to that, the authors reported that the brightness of the system was up to 3.8 × 10^6^ M^−1^ cm^−1^, which was 4-fold higher than the standard NIR-II-emitting nanoprobes. The red-shift was attributed to the quenching of the emissions in NIR-I regions that was caused by the NCs’ aggregation. In particular, it was shown that emissions in the NIR-I region dramatically decreased in emission intensity and lifetime for high-loaded PEM NPs, while emissions in the NIR-II region were slightly affected by the AuNCs concentration inside the polymer NPs. From that, it was hypothesized that the emission in the NIR-I region rose from the surfaces of the AuNCs, which are more affected by the interaction with the other NCs, while emissions in the NIR-II region rose from the inner cores of the AuNCs, since they were less affected by the AuNCs concentration. Such nanoprobes were tested in tissue-mimicking models for in vivo blood vessel imaging in the NIR-II region and they demonstrated promising properties as luminescent nanoprobes for bioimaging with a good signal-to-noise ratio and a depth penetration up to 7 mm. This study elucidated the possibility to obtain NIR-II-emitting nanoprobes, and they proposed the encapsulating polymers as an efficient agent for the tuning of the AuNCs.

In the last years, the encapsulation of emitting AuNCs inside nanocarriers was not limited to AIE, AEE, and DIEE phenomena. Following a different approach, the interaction between photoactive nanoprobes and external stimuli, such as radiations and the chemical environment, can be modulated through encapsulation inside a nanoshell. A well-designed match between the photophysical properties of AuNCs and the ones of the encapsulating agent can be used to develop novel bionanomaterials with superior photophysical properties.

For example, Yao et al. described a “turn-on” NIR-emitting GSH sensor based on AuNCs-loaded silica nanoshells coated with a MnO_2_ layer (AuNCs@SiO_2_@MnO_2_) for the imaging of overexpressed GSH inside cancer cells and tissues [125]. Once encapsulated inside the silica shell, AuNCs exhibited a redshift, with a shift in the maximum emission peak from 610 nm to 652 nm. AuNCs@SiO_2_ photoluminescence was quenched up to 90%, when it was coated by a MnO_2_ layer. This nanoprobe was based on the absorption competition induced emission (ACIE) strategy: the excitation of internal AuNCs overlapped with the absorption bands of the outermost layer of MnO_2_ and, upon irradiation, the excitation light was preferentially absorbed by the MnO_2_ layer, and it induced a quenching of the NIR-emission of inner AuNCs. In addition, AuNCs photoluminescence was recovered in solution by the addition of GSH because it reduced MnO_2_ to Mn^2+^, leading to a disaggregation of MnO_2_ layer around AuNCs@SiO_2_. The nanoprobes demonstrated good biocompatibility inside the cell model. When AuNCs@SiO_2_@MnO_2_ was internalized inside Hela cells and frozen slices of mice tumor tissue, the photoluminescence of the nanoprobes was recovered, and a clear luminescent imaging of the cell cytoplasm was detected, with a penetration depth of 450 µm. This study demonstrated a good match between the photophysical properties of luminescent probe and the coating that could be a useful tool for the design of promising luminescent sensors.

## 3. Perspectives and Conclusions

Even if he applications of AuNCs for bioimaging have been rapidly increasing during the last few years, this family of contrast agents still suffers from some weaknesses, which can be, in principle, diminished or eliminated by a suitable design of the ligands.

Besides emitting in the NIR, a spectral region where the biological tissues are more transparent and autofluorescence is negligible, the PL quantum yield is, in most cases, only moderate (<10%). Although this is enough for their actual application, the level of the signal with respect to the noise S/N could be improved by a rationalization of the design of the ligands. Indeed, a systematic understanding of the effect of the chemical nature of the ligand on the PL quantum yield is still missing, while ligands with a similar structure often affect the PL quantum yield in a very different way. In view of this, a more detailed study on this effect should be carried out in the future, as it would greatly aid in the design of superior probes.

In fluorescence imaging, transparency of the tissues to the excitation light is as important as transparency to the emission light. Nevertheless, in most of the applications, AuNCs excitation is performed with visible light since the excitation of these probes in NIR is less efficient. At the very low concentration regime used for fluorescence imaging, the excitation efficiency is proportional to the molar extinction coefficient, ε, which greatly decreases upon increasing the wavelength. In general, the PL signal, which is detected in defined experimental conditions, is proportional to the brightness, B = ε · QY, where QY is the PL quantum yield. For this reason, controlling the light-absorption properties of AuNCs in order to enhance ε in the NIR is as important as increasing the QY. A powerful but poorly developed approach to enhance the NIR excitation is the use of ligands that are able to absorb NIR light and transfer the excitation energy to the gold core working as an antenna light-harvesting system. This approach should be developed in the future in order to increase the signal-to-noise ratio.

AuNCs’ PL presents quite an unusually long lifetime decay. This feature is often ignored and purely exploited. Indeed, this may represent a big advantage since, in fluorescence imaging, noise is mostly due to excitation, so it can be pulsed and decay very fast, and it is also due to the biological samples’ autofluorescence, which also decays quite fast. Hence, time-gated detection can strongly increase the contrast in AuNCs’ PL-based imaging. In addition, the nature of the ligands is very important since it can affect the PL decay rate. A systematic study of this effect is also still missing, and it would be highly beneficial for the scientific community to carry one out in the future.

In conclusion, during the last years, the scientific community tried to propose novel solutions to the emergent need to find promising luminescent contrast agents that could face, at best, the most updated challenges in biomedical imaging, such as higher brightness and major transparency to the biological tissues. As we highlighted here, recently, NIR-emitting AuNCs were labeled as one of the most promising nanoprobes for the diagnosis of diseases since they have demonstrated to be highly luminescent in the biological window and remarkably biocompatible. Additionally, the design of coating chemicals was selected as one of the most prolific strategies to regulate the physical–chemical, photophysical, and biochemical properties of the NCs. This review reports the most recent developments in the selection and testing of suitable coatings as tuners of the features of NIR-emitting AuNCs and their application in the field of fluorescence bioimaging. Among all, thiolated ligands, biobased molecules, and encapsulating agents were proposed as the most popular classes of capping chemicals for the design of novel NIR-emitting AuNCs as novel nanobiomaterials. So far, brilliant results have been obtained in both in vitro and in vivo testing, and good performances were obtained in fluorescence imaging techniques. However, some major challenges still remain, such as the short-depth penetration of the light emitted by the AuNCs-based nanoprobes and the transposition of these technologies in human-being medicine. For the future, we expect that the selection of highly performative coatings could help us overcome these problems. The hope is that coated NIR-emitting AuNCs can open the door to a bright development in the field of luminescent contrast agents.

## Figures and Tables

**Figure 2 nanomaterials-13-00648-f002:**
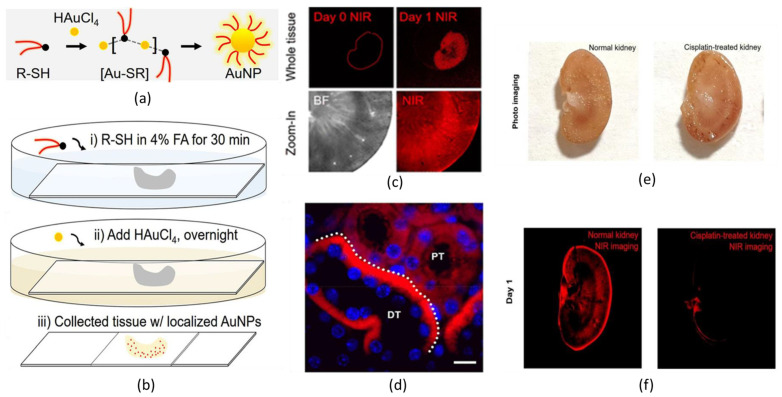
(**a**) Standard synthesis of GSH-AuNCs in solution. (**b**) Synthetic steps to perform precipitation of GSH-AuNCs inside biological tissue. (**c**) NIR fluorescence bioimaging of the whole kidney before (Day 0) and after (Day 1) the precipitation of GSH-AuNCs and zoom-in images of kidney cortex in both bright field (BF) and NIR imaging. (**d**) Confocal fluorescence microscopy imaging of kidney slides shows a preferential precipitation of GSH-AuNCs in the basolateral side of cytoplasm area in the renal distal tubule (DT) rather than in proximal tubule (PT) (Scale bar 10 μm). (**e**) Photos and (**f**) NIR fluorescence bioimaging of normal kidney and diseased kidney (artificially caused by overexposure to cisplatin) after (Day 1) the precipitation of GSH-AuNCs. Adapted with permission from Ref. [69]. Copyright 2019 American Chemical Society.

**Figure 3 nanomaterials-13-00648-f003:**
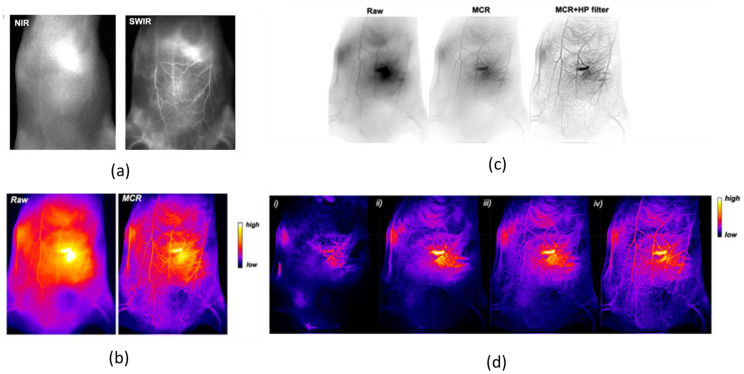
(**a**) Comparison between NIR I bioimaging (λ_exc_ = 780 nm, λ_em_ > 830 nm) and SWIR bioimaging (λ_exc_ = 830 nm, λ_em_ > 1250 nm) of mice ventral area 15 min after injection of (MHA/TDT)-AuNCs. (**b**) Comparison between bioimaging (false colors) of the mouse ventral area before (Raw) and after Monte Carlo constrained restoration (MCR) processing. (**c**) In vivo SWIR bioimaging (reverse contrast) of mice vasculature before image processing (Raw) and after MCR and additional high-pass (HP) filtering. (**d**) SWIR bioimages (false colors) after MCR and HP filtering/processing at (i) 1.5 s, (ii) 5 s, (iii) 25 s, and (iv) 65 s after (MHA/TDT)-AuNCs intravenous injection. Adapted with permission from Ref. [79]. Copyright 2019 American Chemical Society.

**Figure 4 nanomaterials-13-00648-f004:**
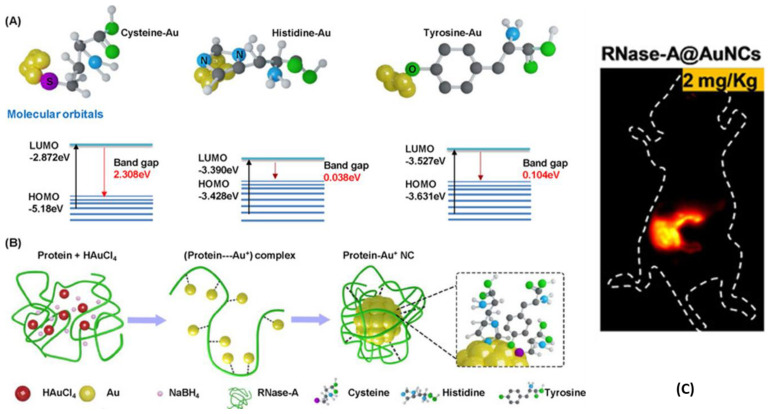
(**A**) Band gaps of gold composites with cysteine, histidine, and tyrosine by DFT calculations. (**B**) Schematic representation of the reaction path for the synthesis of Rnase-AuNCs. (**C**) NIR-II bioimaging (λ_exc_ = 808 nm, λ_em_ > 1000 nm) of mice gastrointestinal region after 2 h post-orally administrated Rnase-AuNCs. Adapted with permission from Ref. [99]. Copyright 2020 Wiley-VCH GmbH.

**Figure 5 nanomaterials-13-00648-f005:**
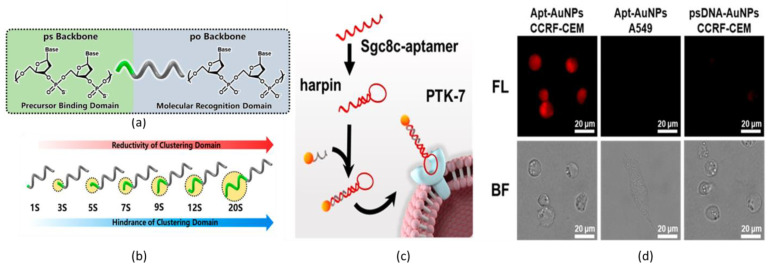
Schematic representations of (**a**) phosphorothioates-modied DNA strand. (**b**) Schematic illustration of trends of the chemical reactivity of DNA strands depending on the phosphorothioates’ unit number in the synthesis of psDNA-AuNCs. (**c**) Schematic representation of Apt@psDNA-AuNCs binding specific PTK7 proteins. (**d**) NIR bioimaging (λ_exc_ = 355–375 nm; λ_em_ = 765–855 nm) and bright field (BF) of A549 cell after 1 h post-incubation of Apt@psDNA-AuNCs. Adapted with permission from Ref. [109]. Copyright 2020 American Chemical Society.

**Figure 6 nanomaterials-13-00648-f006:**
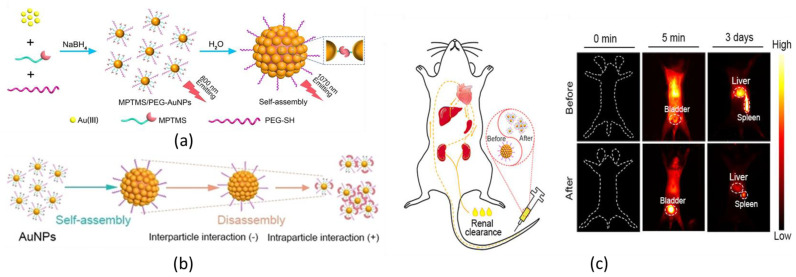
Schematic representation of (**a**) the synthetic route of the MPTMS/PEG-AuNCs and the (**b**) self-assembly/disassembly of MPTMS/PEG-AuNCs. (**c**) In-time renal clearance and liver and spleen retention evaluations by NIR-II bioimaging (λ_exc_ = 808 nm, λ_em_ > 970 nm) in mice after injection of MPTMS/PEG-AuNCs before and after disassembly. Adapted with permission from Ref. [123]. Copyright 2022 Wiley-VCH GmbH.

## Data Availability

Not applicable.

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
