# Peer review of "Luminescent Gold Nanoclusters for Bioimaging: Increasing the Ligand Complexity"

_nanomaterials, 2023, doi:10.3390/nano13040648_

Round 1

Reviewer 1 Report

The review under consideration covers an important piece of modern research in the field of gold nanoparticles preparation, properties, and applications. I loved the idea very much, but I cannot say I enjoyed the manuscript. In my view, it is quite difficult for reading. Some of the aspects listed below are subjective, and I leave the final decision on whether to take my comments into accout to the Editor and the authors.

Also, at this stage I have not included the minor comments - otherwise the list will be too long.

1. The introduction is too vague and fuzzy. It takes about 4 pages, and the main idea is very simple - "AuNCs are advantageous emitters for biological applications, and their properties depend on the preparation conditions". This idea (undoubtedly, scientifically correct) is repeated several times in slightly different context. However, the idea that preparation recipe affects the product properties is so obvious that it does not deserve so many words. If the authors wanted to demonstrate it for a particular case of AuNCs, their Introduction section lacks particular examples (maybe from earlier studies) supporting the idea.

Upon finally reaching section 2, I have had a feeling that I spent months reading about very similar things expressed very generally, without numeric values, chemical formulae, or whatever else making scientific text different from general textbooks.

2. I did not like separation of the introduction into very small sections. For example, sections 1.3 (starting in line 77), another section 1.3 (line 132), and the latter of sections 1.4 (line 142) contain only a single very short paragraph each. In my opinion, the latter two sections repeat the ideas already expressed earlier (in slightly different words) and without any new references/examples. Multiple subsections in the Introduction give the initial impression that the Section 1 is well structured, but these subsections look like some half-finished text.

3. Please verify the references to the figures. The reference to fig 1a in line 53 seems incorrect, since fig 1a does not include any spectra of biological tissues. Figures 2-6 are not referenced in the text at all.

4. Most of the references are just mentioned but not described. There are 159 references in total, and most of them (with very few exceptions) are from 2018 or later. They are very fresh reports which deserve certain discussion in comparison with the earlier 'classical' studies. (Let me also mention that the almost absence of references published earlier than in 2018 leaves a very unpleasant impression that instead of comparative reviewing of recent results as they extend the earlier, better known ones, the authors just made a Scopus search restricting the publication date to 2018-2022 and then constructed the text to somehow fit as many of the query results as possible. This impression is very likely not true, I believe, but I would like to share it).

From the 159 cited references, only ~20 are discussed in certain detail. As an example, lines 159-173 contain 23 references not described any further (less than a line per reference). Just 5 lines 381-385 include 21(!!!) references not mentioned elsewhere in the review. I will repeat to make my idea clear - i) these are very fresh references; ii) they are mentioned in the main part of the review, not in the Introduction (where mentioning of a reference to support a statement without deep discussion is ok); iii) they aim to support the "red line" of the review authors that the diversity of the ligands brings diversity in the properties of AuNCs - but the diversity is not supported by comparative analysis of the cited references. 

As a very rough approximation, 2/3 of the cited references are not described in the review at all, so they just make up the amount of the cited papers but do not bring anything to the contents, which is not appropriate, in my view.

5. I have mentioned that the review is difficult for reading. This is, at least partially, due to too many introductory words. They make the language more 'literary', but at the expence of distracting the readers' attention. I would prefer more academic and laconic style, which would probably give more space for discussing more references. As a representative example, in the paragraph starting in line 406 there are "in fact" thrice (lines 409, 415, 429), "in addition" twice (414 and 431), "furthermore" (419), "interestingly" (421), "thus" (426), "moreover" (432), and "in conclusion" (434). None of these words is bad as such, but when there are too many of them, the text starts to be 'hypnotic' rather than clear and coinsise. 

I suggest the authors to minimise the introductory words throughout the entire manuscript, if they are not necessary to express a particular flavor of the underlying thought.

6. Since the review is based on very fresh references, the 'Future Perspective' and 'Conclusion' sections can be united, in my view, under a common conclusive section. And again, Section 3 contains three entitled subsections, each a single paragraph long, which I also personally did not like (cf. point 2).

Let me also mention that I did not include numerous minor things (such as typos, wrong usage of terms, grammar issues, unexplained abbreviations, etc) in the list above. I hope that many of them will be corrected during the revision of the manuscript.

Author Response

Reply to reviewer 1

The review under consideration covers an important piece of modern research in the field of gold nanoparticles preparation, properties, and applications. I loved the idea very much, but I cannot say I enjoyed the manuscript. In my view, it is quite difficult for reading. Some of the aspects listed below are subjective, and I leave the final decision on whether to take my comments into accout to the Editor and the authors.

Also, at this stage I have not included the minor comments - otherwise the list will be too long.

  1. The introduction is too vague and fuzzy. It takes about 4 pages, and the main idea is very simple - "AuNCs are advantageous emitters for biological applications, and their properties depend on the preparation conditions". This idea (undoubtedly, scientifically correct) is repeated several times in slightly different context. However, the idea that preparation recipe affects the product properties is so obvious that it does not deserve so many words. If the authors wanted to demonstrate it for a particular case of AuNCs, their Introduction section lacks particular examples (maybe from earlier studies) supporting the idea.Upon finally reaching section 2, I have had a feeling that I spent months reading about very similar things expressed very generally, without numeric values, chemical formulae, or whatever else making scientific text different from general textbooks.

Our reply>> Following the suggestion of the reviewer, we shortened significantly the introduction. In particular, we avoided repeating more than once the advantages of AuNC for bioimaging.

  1. I did not like separation of the introduction into very small sections. For example, sections 1.3 (starting in line 77), another section 1.3 (line 132), and the latter of sections 1.4 (line 142) contain only a single very short paragraph each. In my opinion, the latter two sections repeat the ideas already expressed earlier (in slightly different words) and without any new references/examples. Multiple subsections in the Introduction give the initial impression that the Section 1 is well structured, but these subsections look like some half-finished text.

Our reply>> Following the suggestion of the referee, we simplified considerably the introduction. According to her/his comments, we united the introduction eliminating the separation in small subsections. We also checked and modified the text to avoid repeating the same information and concept. We also re-utilized references cited in other sections to avoid having unreferenced discussion.

  1. Please verify the references to the figures. The reference to fig 1a in line 53 seems incorrect, since fig 1a does not include any spectra of biological tissues. Figures 2-6 are not referenced in the text at all.

Our reply>> We apologize for this. Following the suggestions of the referee, we removed the wrong reference of figure 1a. We also cited and added the in-text references of figure 2-6.

  1. Most of the references are just mentioned but not described. There are 159 references in total, and most of them (with very few exceptions) are from 2018 or later. They are very fresh reports which deserve certain discussion in comparison with the earlier 'classical' studies. (Let me also mention that the almost absence of references published earlier than in 2018 leaves a very unpleasant impression that instead of comparative reviewing of recent results as they extend the earlier, better known ones, the authors just made a Scopus search restricting the publication date to 2018-2022 and then constructed the text to somehow fit as many of the query results as possible. This impression is very likely not true, I believe, but I would like to share it). From the 159 cited references, only ~20 are discussed in certain detail. As an example, lines 159-173 contain 23 references not described any further (less than a line per reference). Just 5 lines 381-385 include 21(!!!) references not mentioned elsewhere in the review. I will repeat to make my idea clear - i) these are very fresh references; ii) they are mentioned in the main part of the review, not in the Introduction (where mentioning of a reference to support a statement without deep discussion is ok); iii) they aim to support the "red line" of the review authors that the diversity of the ligands brings diversity in the properties of AuNCs - but the diversity is not supported by comparative analysis of the cited references. As a very rough approximation, 2/3 of the cited references are not described in the review at all, so they just make up the amount of the cited papers but do not bring anything to the contents, which is not appropriate, in my view

Our reply>> Following the suggestion of the referee, we reduced significantly the number of references. Since some reviews already exist discussing NIR photoluminescent AuNCs developed before 2018, we mostly focused our attention on most recent examples. In order to clarify this we added a sentence at the end of the introduction. To reply to the reviewer i) now we clarified why we mainly considered very new references, ii) we strongly reduced the number of undiscussed references, although we maintained references we considered to be interesting for the readers even if not discussed in detail, iii) our idea was to demonstrate that increasing the complexity of the ligand also the complexity and multiplicity of the functions could be extended. We discuss this in pages 9 and 13 in the revised version.

  1. I have mentioned that the review is difficult for reading. This is, at least partially, due to too many introductory words. They make the language more 'literary', but at the expence of distracting the readers' attention. I would prefer more academic and laconic style, which would probably give more space for discussing more references. As a representative example, in the paragraph starting in line 406 there are "in fact" thrice (lines 409, 415, 429), "in addition" twice (414 and 431), "furthermore" (419), "interestingly" (421), "thus" (426), "moreover" (432), and "in conclusion" (434). None of these words is bad as such, but when there are too many of them, the text starts to be 'hypnotic' rather than clear and coinsise. I suggest the authors to minimise the introductory words throughout the entire manuscript, if they are not necessary to express a particular flavor of the underlying thought

Our reply>> According to the suggestion of the reviewer, we eliminated a significant number of introductory words.

  1. Since the review is based on very fresh references, the 'Future Perspective' and 'Conclusion' sections can be united, in my view, under a common conclusive section. And again, Section 3 contains three entitled subsections, each a single paragraph long, which I also personally did not like (cf. point 2). Let me also mention that I did not include numerous minor things (such as typos, wrong usage of terms, grammar issues, unexplained abbreviations, etc) in the list above. I hope that many of them will be corrected during the revision of the manuscript.

Our reply>> Following the suggestion of the referee, we merged the 'Future Perspective' and 'Conclusion' in a single section. We also removed the multiple sections in Section 3.

Reviewer 2 Report

This is well-written review about promising luminescent contrast agents based on Au-nanoparticles. The manuscript could be accepted after minor revision:

1) The authors do no observe a role of Au-Np in enhansment of upconversion luminescence. In my opinion, this is important problem in case of contrast agents. I recommend to complete the review with the upconversion data.

2) The quality of Figure 1 should be increased.

3) The authors should give excitation spectra of luminescence spectra.

Author Response

Reply to reviewer 2

This is well-written review about promising luminescent contrast agents based on Au-nanoparticles. The manuscript could be accepted after minor revision:

1) The authors do no observe a role of Au-Np in enhansment of upconversion luminescence. In my opinion, this is important problem in case of contrast agents. I recommend to complete the review with the upconversion data.

Our reply>> The topic suggested by the referee is very interesting, so we decided to mention it and discuss it in the introduction of the revised version of the manuscript.

2) The quality of Figure 1 should be increased.

Our reply>> According to the referee’s suggestion, we increased the quality of Figure 1. Although the reproduced images are quite old and have no good quality.

3) The authors should give excitation spectra of luminescence spectra.

Our reply>> We agree with the reviewer that the excitation spectra would be interesting but, unfortunately, they are not available in the original publication.

Round 2

Reviewer 1 Report

I sincerely thank the authors for their adequate reaction on my (quite subjective) comments on the initial version of the manuscript.

Upon review of the revised version, I can see that the authors have strongly improved the manuscript to meet the critical points (and also some other minor issues not mentioned in my previous report). I still cannot say that I enjoy the text to recommend it to my students as an ultimate and comprehensive source of information on recent research in the field of fluorescent Au nanoclusters, but I have to establish the fact that the collected data and discussion (although not meeting my personal expectations) is a strong and important contribution to a very important field of modern chemistry.

I suggest publishing the manuscript, possibly accounting for several minor comments below.

1. In lines 68-70: "Despite the most recent developments, such emitters often have not shown brilliant performances, like low quantum yield [26], scarce photostability [27], photobleaching [28], or insufficient biocompatibility [29].

This text is somewhat misleading, since "brilliant performances" (although "not shown" are announced, but then the drawbacks are listed. I would suggest putting this sentence into more straightforward form, something like "... emitters have sometimes suffered from certain drawbacks, such as ..."

2. Line 77. I suggest changing "determinate" to "determine".

3. Line 89. The HOMO and LUMO abbreviations already contain "molecular orbitals", so "HOMO and LUMO molecular orbitals" seems not perfectly correct (although fully understandable).

4. I understand the authors' point that they used only very recent original reports in view of existence of reviews of the earlier papers. However, I would suggest specifically adding the references to few of such reviews (these suggested for further exploration by the authors of the present manuscript). I think, the last sentence in lines 114-115 is a proper place to give the references. If the authors do not want to ruin the subsequent numbering of the references, I suggest putting a footnote with direct links to the reviews. 

This point may seem too much work, but I really find it important to enable the readers to follow the back references.

5. In line 170, a "reductant-free" synthesis of AuNCs is mentioned. However, it is clear that Au(III) as well as Au(0) cannot be converted into Au(0) without a reducer. If I have understood correctly, this statement means that under the action of MW radiation, GSH or other species which normally do not induce reduction into Au(0) start acting as more efficient reducers to afford the AuNCs. However, i would avoid the term "reductant-free" as it seems chemically misleading (it could mean, for example, that Au(I) complexes become fluorescent without reduction).

6. In line 343, I would reconsider the section title ("Bio-based ligands"), since the thiol-based ligands described in the previous section are bio-based as well, and on the other hand some of the bio-based ligands mentioned in section 2.2 contain thiol groups as well. I cannot recommend a proper title at once, though.

7. Line 436. I would add chemical composition of the cyclic peptide, if it is known from the cited reference.

Apart from these really minor things, I think that the revised manuscript should be accepted, even though I have not 100% enjoyed the style.

Author Response

Reply to Reviewer 1

Comments and Suggestions for Authors

I sincerely thank the authors for their adequate reaction on my (quite subjective) comments on the initial version of the manuscript.

Upon review of the revised version, I can see that the authors have strongly improved the manuscript to meet the critical points (and also some other minor issues not mentioned in my previous report). I still cannot say that I enjoy the text to recommend it to my students as an ultimate and comprehensive source of information on recent research in the field of fluorescent Au nanoclusters, but I have to establish the fact that the collected data and discussion (although not meeting my personal expectations) is a strong and important contribution to a very important field of modern chemistry.

I suggest publishing the manuscript, possibly accounting for several minor comments below.

  1. In lines 68-70: "Despite the most recent developments, such emitters often have not shown brilliant performances, like low quantum yield [26], scarce photostability [27], photobleaching [28], or insufficient biocompatibility [29]." 

This text is somewhat misleading, since "brilliant performances" (although "not shown" are announced, but then the drawbacks are listed. I would suggest putting this sentence into more straightforward form, something like "... emitters have sometimes suffered from certain drawbacks, such as ..."

Our reply >> We changed the text in the revised version of the manuscript according to the suggestion of the referee.

  1. Line 77. I suggest changing "determinate" to "determine".

Our reply >> We would like to thank the referee for the correction. We modified the text accordingly.

  1. Line 89. The HOMO and LUMO abbreviations already contain "molecular orbitals", so "HOMO and LUMO molecular orbitals" seems not perfectly correct (although fully understandable).

Our reply >> We agree with the comment of the referee. We modified the text according to his suggestion.

  1. I understand the authors' point that they used only very recent original reports in view of existence of reviews of the earlier papers. However, I would suggest specifically adding the references to few of such reviews (these suggested for further exploration by the authors of the present manuscript). I think, the last sentence in lines 114-115 is a proper place to give the references. If the authors do not want to ruin the subsequent numbering of the references, I suggest putting a footnote with direct links to the reviews. 

This point may seem too much work, but I really find it important to enable the readers to follow the back references.

Our reply >> We would like to thank the referee for his comment. In the revised version of the manuscript, we modified slightly the text adding other reviews that have been done on the topic in lines 114-115.

  1. In line 170, a "reductant-free" synthesis of AuNCs is mentioned. However, it is clear that Au(III) as well as Au(0) cannot be converted into Au(0) without a reducer. If I have understood correctly, this statement means that under the action of MW radiation, GSH or other species which normally do not induce reduction into Au(0) start acting as more efficient reducers to afford the AuNCs. However, I would avoid the term "reductant-free" as it seems chemically misleading (it could mean, for example, that Au(I) complexes become fluorescent without reduction).

Our reply >> We would like to thank the referee for his comment, we eliminated the term “reductant-free” according to his suggestion.

  1. In line 343, I would reconsider the section title ("Bio-based ligands"), since the thiol-based ligands described in the previous section are bio-based as well, and on the other hand some of the bio-based ligands mentioned in section 2.2 contain thiol groups as well. I cannot recommend a proper title at once, though.

Our reply >> We would like to thank the referee for his comment. We modified the title of section 2.1 adding also a sentence (page 4) to clarify the division of the two sections. 

  1. Line 436. I would add chemical composition of the cyclic peptide, if it is known from the cited reference.

 Our reply >> We added the peptide sequence according to the suggestion of the reviewer.

Apart from these really minor things, I think that the revised manuscript should be accepted, even though I have not 100% enjoyed the style.